# Perspectives on Quality Risk in the Building Process of Blue-Green Roofs in Norway

**Erlend Andenæs \*** , **Atle Engebø, Berit Time, Jardar Lohne, Olav Torp** and **Tore Kvande**

Department of Civil and Environmental Engineering, Norwegian University of Science and Technology, 7491 Trondheim, Norway; atle.engebo@ntnu.no (A.E.); berit.time@sintef.no (B.T.); jardar.lohne@ntnu.no (J.L.); olav.torp@ntnu.no (O.T.); tore.kvande@ntnu.no (T.K.)

\* Correspondence: erlend.andenas@ntnu.no

**Abstract:** As climate change brings an increase in torrential rain events in Nordic climates, new technologies are developed to manage stormwater. Blue-green roofs are constructed as a means to reduce the runoff of stormwater from roofs and reduce the risk of urban flooding. However, compared to conventional roofs, blue-green roofs represent different construction and operation conditions, which may affect the long-term integrity of the roof. The purpose of this research is to understand the variety of perspectives on how different actors perceive and manage quality risks related to blue-green roofs—that is, the probabilities and consequences of defects. The quality risks of blue-green roofs have been investigated through document studies and interviews with actors in the Norwegian building sector. Data have been collected from actors across the building sector to map differences in how risk is managed from several perspectives. The findings show that actors view quality risk in very different ways. While building owners are primarily concerned with the quality of the finished product, the primary concern of other involved actors may be to ensure that eventual defects cannot be attributed to their own activities. The efforts of the various actors to reduce the risks in their own activities may not necessarily reduce the risk of defects in roofs. To ensure a more comprehensive management of quality risk in blue-green roofs, it is necessary to consider the perspectives and incentives of all involved actors. This way, a framework could be developed as a feasible tool in blue-green roof projects.

**Keywords:** risk; blue-green roof; building process; quality risk; climate adaptation

## 1. Introduction

The densification of cities causes an increasingly large fraction of the ground surface to be covered by impermeable materials, leading to a greater risk of stormwater flooding [1]. In certain climates, this risk is exacerbated by climate change, for instance, in Norway, where an increase in torrential rain events is forecast in the future [2]. To a greater degree than ever, it is necessary for cities to address the threat of urban flooding through effective stormwater management. In addition to the threat of urban flooding, buildings also need to be made more resilient in general to face the challenge of a changing climate [3,4]. In sum, the future climate requires a better understanding of risk and how to handle risk in the built environment. In this paper, the investigated risk perspective is that of quality risk, the term being understood as "the likelihood and consequences of building defects occurring".

A blue-green roof is a roof assembly where rainwater is stored using plants and various substrate layers. The difference between an ordinary green roof and a blue-green roof is that the latter is purpose built for stormwater management purposes. This might include a larger water storage capacity than what is needed for the plants to survive [5]. Blue-green roofs are found to have considerable potential

for stormwater control within a building site, both when applied in new buildings or retrofitted onto existing buildings [6].

In practice, the reasons why blue-green roofs are chosen in a given project can be divided into two categories: those instances where the initiative is taken by the project owner, or those where blue-green roofs are built to satisfy external (i.e., legal) requirements. Until recently, green roofs have tended to be built for reasons related to the former category. They were typically considered a novelty and chosen for aesthetic reasons, to add an element of greenery to a building [7–9]. The stormwater management properties of the roofs were largely seen as an optional bonus and rarely considered in the stormwater management plan (although this has been a motivation in certain projects). However, in recent years, new requirements for stormwater management have become stricter in Norway as well as many other countries. International research has documented the benefits of using green roofs to reduce floor risk in urban areas [10–13]. In Norway, rules on a regional or municipal level are mandating the use of solutions for detention, retention, and local infiltration of stormwater, as shown in [14,15]. Green and blue-green roofs see increased interest as a measure for stormwater management in urban areas [6]. This could lead to an increased use of green roofs in building projects, without an owner-driven initiative for their construction. As blue-green roofs become more common, it is vital to understand the risks they may pose for the buildings on which they are built.

Blue-green roofs are commonly built on top of a conventional, compact, flat roof assembly [16]. Many of the risk factors associated with blue-green roofs will also be relevant for compact flat roofs. Therefore, this article will also include risk factors for compact flat roofs, as the volume of literature available on these roof assemblies is substantially greater than that on green or blue-green roofs.

It is known from experience that roof defects are a recurring problem in the building stock today [17]. Comprehensive, quantitative data on the prevalence of building defects are, however, not available [18]. Attempts to establish a common Norwegian national database for building defects have so far failed. Research suggests that the most commonly investigated roof defects in Norway concern the intrusion of rainwater or snowmelt into the building structure. Gullbrekken et al. [19] found that roof defects comprise 22% of building defect cases in Norway, with precipitation damage occurring in 51% of cases of flat roof defects. Moisture damages (all moisture sources) accounted for 89% of all investigated defect cases for flat roofs. Moisture compromises the insulating capabilities of building insulation, may stain materials, and facilitates the growth of fungi.

It is specified in the Norwegian technical regulations for buildings that moisture intrusion must not occur in such a way that the building may be damaged [20]. The requirements are function-based and independent of the solutions chosen to meet them [21]. This approach gives designers wide freedom to choose a solution, but also increases the room for error. A means to reduce the risk of error while retaining the freedom is therefore highly desired.

A building project involves several actors, each responsible for a share of the final product. For large building projects, the organization may be very complex. It is not always clear to everybody where the borders of responsibility go between actors. Additionally, actors may perceive risks and challenges differently. This may potentially create borderline cases where defects occur because nobody considered the quality risk of the chosen solution. A clear mapping of the overall risk picture for quality defects is therefore required, including collecting the perceptions of risk from the various involved actors. In light of this complex problem, this article investigates the following research questions:

- How do the various actors in the building process perceive the risks of blue-green roofs?
- How are the risks associated with blue-green roofs currently managed?
- How can the management of quality risk be improved?

The following limitations apply to the research: only the perspectives of certain actors involved in a building project are investigated, further detailed in the Methods chapter. Natural hazards are not included. Positive risk (beneficial uncertainty) is not considered. The investigated time period is

between conceptualization and handover of the building, excluding the operations/maintenance phase or end-of-life.

## 2. Theoretical Framework

### 2.1. Blue-Green Roofs

Blue-green roofs are roof assemblies wherein a mat of vegetation and its substrate layers are used to store precipitation water, making the roof part of a stormwater management strategy. Any green roof built for this purpose can be considered a blue-green roof [5]. In Norway, climate change is expected to cause an increasing frequency of extreme precipitation events [2], which may lead to flooding in urban areas. Using roofs to manage stormwater is an important part of the strategy to combat urban flooding [6]. A blue-green roof will add retention capacity (evapotranspiration of water) to the detention capacity (temporary storage of water) provided by conventional green roofs [12,13].

A green or blue-green roof assembly consists of various layers, typically constructed on top of a conventional compact roof structure [22,23] (Figure 1). Plants, commonly sedum or other succulents, grow in a substrate and form the outer roof surface. A sheet of geotextile separates the substrate from the water storage layer, which commonly consists of extruded plastic boards or crushed Leca. The water storage layer is designed not to give a standing water pressure against the roof membrane, by storing water in cups, boxes, or capillary pores. The layer also provides drainage for excess water. A root barrier is applied against the roof membrane to protect it from root damage; this may also be achieved through chemical treatment of the roof membrane itself. Optionally, the plants and their substrate can be replaced with a permeable pavement, to create a so-called "blue-grey roof" [24].

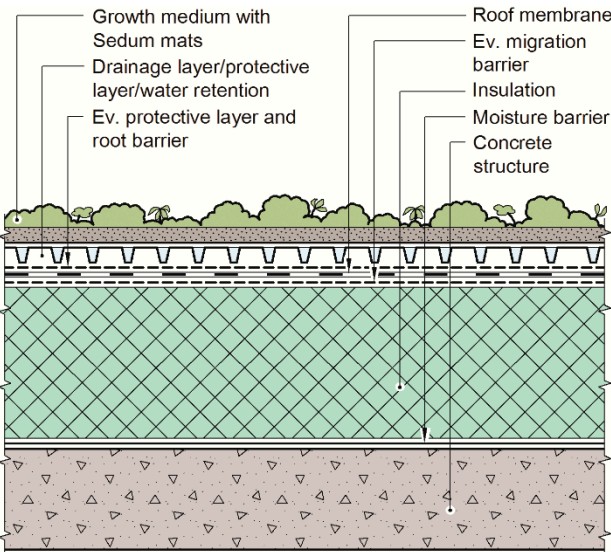

**Figure 1.** Example of a blue-green roof assembly. The thickness of the growth medium may be vastly increased for intensive green roofs. The water storage and drainage layer shown is based on extruded plastic boards, but this layer may also consist of gravel or crushed baked clay. Alternate concepts for blue-green roof assemblies also exist, see, for instance, Ref. [12]. Illustration ©SINTEF.

Adding additional layers onto the roof assembly will change the physical operating conditions of the roof membrane, which is the watertight layer that keeps moisture out of the building envelope. Parameters such as moisture, temperature, solar irradiation and mechanical pressure will be very different for a roof membrane lying underneath a blue-green roof, compared to exposed roofing. In certain aspects, these conditions can be beneficial for the long-term integrity of the roof, particularly the reduced temperature fluctuations and solar irradiation [23,25,26]. As such, the economic benefits of green roofs have been the focus of much research. However, the building technical aspects of

green roofs have not been well studied in research literature [5]. The physical operating conditions of blue-green roofs are very different from conventional roofs, with associated risks that must be managed. Crucially, the roof membrane will not be available for inspection after the green roof is constructed, and repairs to the roof will subsequently be vastly more expensive and difficult than is the case for conventional roofs. If there is a defect in the water-proofing layers, it is likely not to be discovered until water has penetrated the whole roof and soiling can be seen inside the building. By then, the defect may have caused significant damage, which is expensive to repair, and it may be difficult to find. However, if the roofing layer is intact when the roof is finished, it is likely to remain intact as the blue-green layers offer some protection from weather and wear. It is therefore of vital importance that the roof is properly designed and constructed, and that its integrity is secured throughout the construction period. Risk factors threatening the quality of the roof must be mapped and made known so the risk can be reduced.

### 2.2. Risk and Quality Risk

Risk is commonly understood to mean the negative consequences of uncertainty, that is, the probability of negative events and their consequences. A more formal definition suggests that "Uncertainty is an event that, if it occurs, has a positive or negative effect on a project's objectives" [27,28]. There are several types of risks, used in different contexts. It is the impression of the authors that risk management literature tends to focus on risk in terms of physical hazards, schedule delays or cost overruns. Taroun (2014) sums up the traditional risk perception in the construction industry as "the variance of cost and duration estimation" [29]. Quality can be defined as "meeting the legal, aesthetic, and functional requirements of a project" [30]. "Quality risk" is a term here used to describe the likelihood and consequences of building defects occurring, rather than the effects of defects on the project's schedule and cost.

However, the term "quality risk" is not well defined in literature. Other terms found to describe the same subject include "defect risk" [31,32], "quality management" [30], "quality deviations" [33] or "defect management" [34]. In the following, we use "quality risk" to include all of these terms. Defects include design flaws, build flaws, material flaws, accidental damage, gradual degradation, and use flaws. The latter two categories of defects occur during the use phase of the building (barring exceptionally long construction periods) and are excluded from the scope of this article. It has been estimated that defects account for 2–6% of the cost of production of a building [35].

Quality risk is a type of risk whose primary consequences are usually restricted to the building itself, or to its occupants in rare cases of catastrophic failure. Although the costs associated with quality failure can in some cases be divided, shared, or shifted onto actors in the construction process [27], the defects themselves and their consequences cannot be taken out of the building. In a sense, the building itself is the primary stakeholder when it comes to quality risk. For all the other involved parties, quality risk must be seen in relation to financial risk. The reduction of quality risk is a benefit that will have to be weighed up against its costs. Most of the parties involved in the building process are not likely to be directly impacted by building defects but will instead incur costs of repairs and/or compensation for defects for which they are found responsible. It follows that there are two ways for an actor to manage quality risk: avoiding defects or avoiding responsibility.

### 2.3. The Building Process

A conceptual illustration of the building process and its main involved actors, exemplified in a design–build (DB) delivery model, is shown in Figure 2. As the figure shows, actors may be directly involved in the production organization of the specific project, attached in a more peripheral fashion in the supply chain, or influencing the project by laying premises or legal frameworks. Different actors are involved in different phases of the project, and there may not be direct communication between all the actors. However, decisions made by one actor early in the process may influence other actors in later phases. For instance, contractors build according to plans submitted by designers. Likewise, premises

and conditions set by actors involved late in the process may influence the decisions made in the early phases. For instance, materials chosen by the architect need to be available for the supplier to deliver.

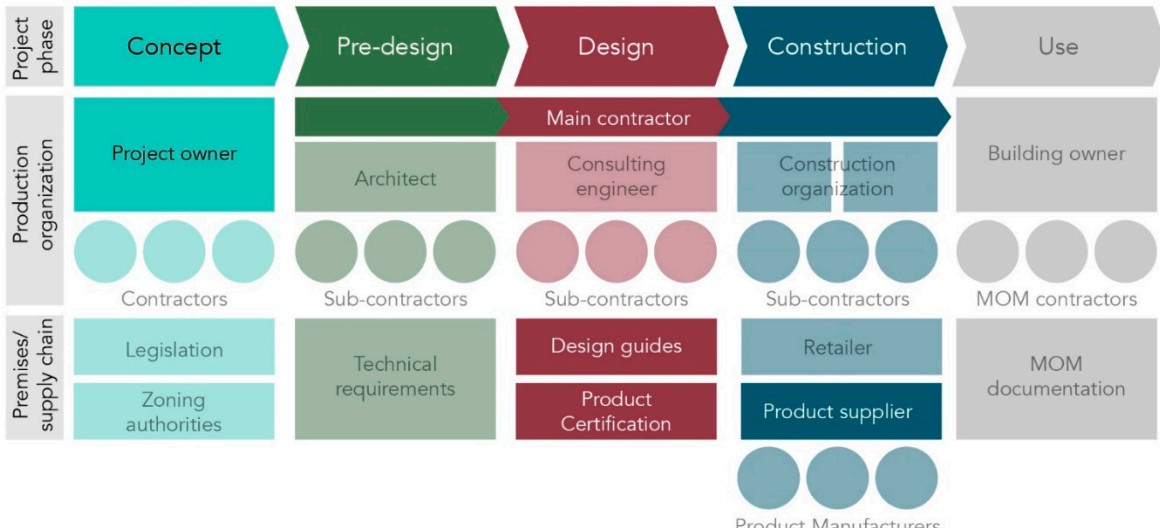

**Figure 2.** Illustration of the building process and its main involved actors in a design–build (DB) contract delivery model. Actors and phases considered outside the scope of this article are shown in faded colors. The inclusion of actors in this article represents all project phases until the point of handover, and actors both within and outside the production organization. The split boxes under "construction organization" denote a larger organization with more sub-contractors than what is common in the other phases.

## 2.4. Actors in the Building Process

The sections below outline the main actors in the building process and their general roles in a project, as shown in Figure 2. While there may be variations depending on the delivery model for the project, the tasks performed by each actor do not usually differ significantly from what is listed [36].

### 2.4.1. Project Owner

The project owner initiates and conceptualizes the building and will in many cases end up owning it upon completion. The project owner has a governing task of deciding the mission, goals and organization, and a supporting task of providing resources and enabling formal decisions [37,38]. Defects that occur during the operations phase, or that are discovered after the contract warranty period, will be the responsibility of the building owner, and as such the owner carries the greatest quality risk. The long-term integrity of the roof is therefore a key point of interest for the building owner. As the owner also initiates the project, they will be able to influence the roof concept to a significant degree through the tender process, by choosing pre-design specifications (beyond what is mandated by technical requirements), or through contract design.

### 2.4.2. Designer/Architect

One of the main functions of the design phase is for the owner to communicate to the designers his or her needs and objectives in initiating the project [39]. For a construction project, the designers are the architects and engineers responsible for detailed design. They are responsible for transferring theoretical and practical expertise into the building project, to ensure that the chosen design conforms to all relevant regulations and standards. The designer and/or architect carry the responsibility for design flaws, so it is in their interest to ensure that their recommendations adhere to the best possible practice.

One way to document best practice is to anchor the recommendations in design guide documents issued by third-party advisory bodies. These may be governmental bodies, academic institutions,

industry organizations, or independent associations. In Norway, the research organization SINTEF develops building design guides through its subsidiary SINTEF Community. The design guides are widely used and enjoy a good reputation throughout the Norwegian construction industry [40,41]. The solutions are based on the best practice from the construction industry and independent forefront research.

### 2.4.3. Contractor

The role of the contractor depends on the type of delivery model applied in the project in question. The most common delivery models in Norway are design–bid–build (DBB) and design–build (DB). In a DBB model, the contractor is engaged by the project owner after the building has been designed by architects and engineers. The contractor will then construct the building according to the given plans and blueprints. Moreover, in DB contracts, the design responsibility is delegated to the contractor, as shown in Figure 2. The contractor is engaged after the owner has drawn up pre-design requirements for the building, whereupon the contractor is responsible for the design process as well as constructing the building [42]. For the owner, the change from DBB to the DB model will mean less liability towards the contractor for design documents. As a result, the owner's risks, and, consequently, the potential contractor's claims, will in theory be mitigated substantially [43].

Making constructability knowledge accessible to the designer(s) and/or architect(s) and at an appropriate level of detail at the right time in the design process is a significant opportunity to improve the constructability of design [44]. Therefore, to mitigate quality risk in the building process of blue-green roofs, the contractors should be allowed to contribute with their constructability expertise in the design process of the project. Methods for employing the full potential of constructability expertise from all sources (including specialty contractors) exist in the use of integrative/collaborative mechanisms [45]. This may be achieved in practice through implementing a collaborative project delivery method that seeks to create an effective integrated team through early involvement of the contractor along with a team with the right expertise seeking to take full advantage of the team's collective 'knowledge pool' [42,46].

### 2.4.4. Material Supplier

As a construction project is a temporary construct that produces a one-off product, its construction supply chain is characterized by instability, fragmentation, and by the separation between the design and the construction stage [47], Typically, products are chosen by the contractor based on recommendations from the design phase. The supplier does not typically participate in a project organization, but the performance of their products needs to match the documented specifications. Defects in materials occur when the operating conditions they are subject to exceed the performance limits of the material. This can occur through improper use of the material, if the operating conditions are more extreme than anticipated, or if the performance limits of the materials are lesser than documented [48]. In the latter case, defects will be the responsibility of the material suppliers. To reduce the risk of defects, suppliers therefore aim to document the performance of their materials as accurately as possible. The performance parameters of materials are determined by methods specified in industry testing standards. The material supplier may also reduce the risk of improper use through supplementary product documentation, such as installation manuals or maintenance plans, or by training workers.

It is also possible for a supplier to enlist a third-party body to independently verify the product's performance and compliance with the building code, and overall assess its general suitability as a construction product. The product can then be certified with a technical approval if it meets the criteria. A technical approval is additional documentation beyond what is required for CE-marking (European standards conformity marking). Institutions offering such certifications include SINTEF (Norway), RISE (Sweden), TÜV (Germany) and BBA (UK). Certifications include testing according to the aforementioned standards, as well as independent assessment of the product's properties [49].

However, when a material property is declared using an industry standard, for instance, dimensional stability or tear strength, the declared performance limit is not necessarily universal. The given property represents the performance limit of the material when subject to the test specified in the standard. While standardized tests aim to reflect realistic use cases, this may not always be possible in practice. For instance, the testing standard for root penetration through roofing membranes presumes one specific plant that may not be commonly found on blue-green roofs [50].

2.4.5. Other Actors

Aside from the main actors listed above, a handful of other actors can be said to be involved in a building project, if only in a tertiary fashion:

National construction authorities shape and enforce national building regulations, which also yield a large influence on the building. These technical regulations concern structural safety, universal accessibility, energy requirements, fire safety, and other technical requirements the building must conform to [51].

The local construction and planning authorities give the premises for initiation of the project. Local laws may yield great influence on the chosen solutions for the building, primarily through zoning regulations that govern matters such as the building's height, footprint, or placement, but also the building's connections to local infrastructure. For instance, local stormwater management practices may demand that roof runoff water is infiltrated into the soil within the borders of the property [14]. Such a requirement makes internal drains unfeasible, but external drainage solutions from flat roofs may have issues with snowmelt re-freezing. This balance between stormwater management and building physics is currently a challenge under further investigation [52].

## 3. Methods

The research has been conducted in several phases, described separately in [53–56]. The methodology includes interviews with actors from several parts of the Norwegian building industry, searches in the national database for building tenders in public construction projects, and two document studies. The results of each study phase are here compared to form an overall impression of risk management of blue-green roofs in the Norwegian building sector. As the building process is complex and involves more actors and perspectives than could reasonably be studied in full given the available time and resources, it was decided to single out a limited number of actors for further study. It was decided to focus on the project owner, contractor, supplier, and design basis guidelines to get representation from each step of the building process as illustrated in Figure 2. The selected perspectives include actors within the organization of an individual building project, as well as actors that influence the project without (necessarily) participating in it. Every phase of the construction project until the point of handover is represented.

### 3.1. Interviews—Overview of the Problem

The first phase of the research was focused on collecting data on risk elements regarding blue-green roofs. As they are usually built upon compact, flat roofs, they share many of the same risk elements. It was decided to find out to what degree defects in compact roofs were occurring in Norway. However, comprehensive quantitative data on building defects in Norway do not exist. Certain companies in the insurance industry, or some advisory firms, maintain their own databases based on cases the company has been involved in. However, these databases are not synchronized and not comprehensive. There exists a mandate for the National building council to create a national database of building defects, but it is yet to materialize [18].

As such, qualitative data had to be gathered instead. A qualitative approach appears to be a common method to study building defects, as quantitative data is not available. Examples are found in [35,57]. To map which types of roof defects are the most prevalent in Norway and how they occur, interviews were conducted with experts on compact roofs from the Norwegian building sector. A total

of 7 people were interviewed in 5 separate interviews, comprising property developers in the public sector, an insurance agent, a representative from a government agency, and a materials supplier. Limiting the number of interviews allowed for a deeper analysis of the contents of each interview. Thus, the emphasis was on conceptualization through generating a richly textured understanding of experience rather than seeking to "frame" or "contextualize" the sample size [58,59]. The interviews were carried out over the phone or in person. An interview guide was developed and made available to the interviewees prior to the interviews. The guide helped structure the interviews, which were designed to be loose to allow a natural flow of conversation. The interviews were recorded and transcribed, so as not to interrupt the conversations.

### 3.2. Public Tender Database—Project Owner Perspective

This research phase aims to map to what degree the owners in public projects are making use of their influence in the concept/pre-design phase. To investigate how project owners manage risk in green roof projects, it was decided to examine the specifications given for green roofs in construction tenders. The Norwegian public tender database, Doffin [60], was searched for mentions of the phrase "green roof", yielding four results from recent projects that were further examined. Additionally, the building division of a municipality near Oslo was contacted to obtain the pre-design reports for known construction projects that included green roofs; this yielded a further three results.

The seven project tenders were examined to determine the contract design, the intention behind building a (blue-) green roof, and the relevant project phase. Where technical documents were available, the level of technical specifications in documents was investigated. This included, e.g., the type and placement of the roof membrane, the stormwater management function of the roof, the location and type of drains, references to roofing integrity, or other mentions of specific risks. This made it possible to assess the overall thoroughness of risk management from the project owners' side, in the phase where they yield the greatest influence over the project.

### 3.3. Material Supplier Datasheets—Supplier Perspective

Risk management on the part of the material supplier was examined through an investigation of product declarations and documentation, specifically for roofing membranes. The research phase aims to chart the documentation required to fully understand the characteristics of a given product, to assess its suitability for a given construction project. Several products available on the commercial market were singled out for study through data sheets, assembly instructions, and, where available, technical certifications from third parties. Documents were searched to map which standards the products conform to, describing how the material properties were determined.

### 3.4. Building Design Guides—Advisor Perspective

The design of buildings in Norway is greatly influenced by design guidelines developed and published by SINTEF Community, formerly the Norwegian Building Research Institute. More than 800 design guides exist, covering every phase of a building's lifetime and every part of the building structure. Issuing such detailed recommendations gives SINTEF Community a certain level of responsibility in building defect cases, putting them at risk of receiving the blame if a recommended solution turns out to be faulty. For this reason, the recommendations in the building design guides are periodically thoroughly reviewed.

To investigate how the SINTEF Building Design Guides manage quality risk in practice, 9 design guides pertaining to green roofs and compact roofs were examined. There are four levels of recommendations in the design guides, ranked by decreasing strictness as follows:

- Required by law, mandatory (i.e., fire safety measures).
- Strongly recommended (i.e., moisture safety measures)
- Recommended (i.e., measures for building longevity)

- Optional (i.e., aesthetic measures)

The number of recommendations of each level in the nine design guides was counted. Explicit mentions of quality risk issues in the design process were also noted, to count how directly quality risk is considered in the Building Design Guides. A small literature search was also conducted to examine information overload in a design project and the limit of effective information processing in the human brain.

### 3.5. Project Delivery Methods—Contractor Perspective

The management aspects of flat-roof construction were studied qualitatively, described in Section 3.1 [54,61]. While not being examined directly, the project delivery methods have been shown to have an immense effect on how risk is perceived, managed, and allocated in projects. The concept of collaborative project delivery methods was examined in a separate scoping review, assessing 156 articles concerning Partnering, Integrated Project Delivery, Alliancing, Relational Contracting, and Relationship-Based Procurement [62]. The role of the contractor has also been examined empirically through case studies [42,46].

## 4. Results

### 4.1. Summary of Main Findings

A summary of the main findings of the study presented, through the perspective of actors shown in Figure 2, is given in Table 1. In the table, the advisory body and product certification function have been merged.

**Table 1.** Summary of the main findings. The column "Identified measures" refers to measures discussed in the Discussions chapter, Section 5.3 of this article.

| Actor | Examined Project Phase | Risk Avoiding Factors | Risk Management Factors | Identified Measures |
|---|---|---|---|---|
| Project owner | Concept | • Delivery below expected quality<br>• Design flaws | • Specifications in tender documents<br>• Contract delivery model | • Cooperative delivery models<br>• Demands of contractors<br>• More detailed specifications in tenders |
| Main contractor | Pre-design, design, construction | • Exceeding budget/schedule<br>• Defects occurring on site | • Coordinating actors on site | • Delivery models<br>• Structured production organization |
| Advisory body (design guides, certification) | Design | • Design flaws<br>• Material flaws | • Design guides<br>• Certification documents | • Establishing risk hierarchy<br>• Stratifying design guidelines |
| Supplier (product data sheets) | Construction | • Material flaws causing building defects | • Product performance declarations | • Practical instructions<br>• Participation in project organization |

### 4.2. Interviews

When interviewed about the nature of roof damages, respondents mentioned challenges both on a physical and processual level. A general trend in the interview responses was the observation that complex roofs are more challenging than simple surfaces. Complex geometries, corners, transitions between building elements, and perforations of the roofing for technical equipment were all seen as challenging to work with and prone to defects. When a building is expanded and the new roof is joined to the old building, it is difficult to verify the integrity of the seam. It was also observed that material defects appear to be a rarity, with materials generally delivering on their specifications. Improper design or installation is a more common cause of defects. Design errors and build errors were thought to be equally common. Counterfeit or sub-standard materials (CFSS) were not considered common in Norway. Only two of the respondents had ever heard about cases involving CFSS. Fraudulent workmanship is thought to be a bigger problem than fraudulent materials. However, the subject is not given much attention and it may be a more common phenomenon outside the professional market.

On a processual level, the respondents stressed the challenge of cost versus quality. A system of technical approvals for materials is used to certify compliance with standards and to document that a product is usable in a Norwegian context. However, there appears to be a sentiment that any product with a technical approval is as good as any other, so builders tend to choose the cheapest option if several are available, regardless of their technical specifications. However, technical approval does not automatically mean that the product is suitable in a specific project. For instance, wind loads vary greatly between locations, and roofing may be torn off if it is not rated for the design wind loads.

Building owners generally expressed a large amount of trust in their contractors. Large, public building owners may have long-term agreements or partnerships with construction companies for construction and maintenance of their buildings. This is a measure to save cost, but also reduces the risk of fraudulent workmanship in the project. Construction companies may also partner with suppliers, creating a chain of agreements between large, professional actors in the building sector. However, in the "consumer construction market", between smaller and less professional actors such as homeowners or small businesses, relations may be less formal and less anchored in solid contracts. It is, however, challenging to acquire an overview of this sector.

Respondents also noted that errors in the use phase could lead to roof defects. Typically, this included flawed maintenance or lack thereof entirely. However, the use phase is to be considered outside the scope of this article.

### 4.3. Public Tender Database

The majority of the case projects (five out of seven) concerned calls for design–build contracts. Little consistency was observed in terms of risk management in the available documents. In three of the cases, a green roof was only mentioned as an option in the contract tenders, with no further technical specification. This leaves the design of the blue-green roof to the contractor, without additional input from the owner.

Where technical documents were available (four of the seven cases), the level of given specifications was not consistent. Pre-design reports mentioned the green roof in all cases, but the level of detail varied between them. Only two pre-design reports specified the design of the roof layer. Only one report recommended that the roof undergo an integrity test before the green roof was assembled. References were found to further literature (SINTEF Building Design Guides [22]) in several cases, but the guides do not necessarily cover special use cases such as transitions between building elements.

The thoroughness of both the tenders and the pre-design reports appears to depend entirely on the persons who wrote them. There does not seem to be a framework to follow when specifying technical details on this level of the building process, where owner input has the greatest possible influence on the finished product. As a result, the application of this influence by public project owners is inconsistent at best and absent at worst.

### 4.4. Product Datasheets

The performance declarations of the investigated roof membranes listed different parameters according to 18 different standards; however, not all standards were used by any one product. One product declared properties in accordance with 13 standards, others used as few as seven. These standards are in turn referencing other standards. The standard EN 13707:2013 Flexible sheets for waterproofing–Reinforced bitumen sheets for roof waterproofing–Definitions and characteristics [63] lists 23 other EN and ISO standards as "indispensable for its operation". Additional documentation was also available through product certification and assembly instructions.

This research phase concludes that an understanding of a product's performance parameters and hence an assessment of its suitability for use in a single project requires in-depth expertise or a significant investment of time and resources. Access to all the 18 standards used to declare performance was found to cost upwards of NOK 8000 (around EUR 800) combined, making a full assessment of the product's properties a costly affair as well as a time-consuming one.

It was also found that technical approval of products is well established in the Norwegian market and generally trusted by all the enquired actors. While the certification process is voluntary to participate in, project owners often require that only materials with a technical approval may be used in projects. As such, certification may be regarded as a requirement for a product to be competitive on the market.

In the interview phase, however, some respondents noted that technical approval alone is no guarantee against defects, as operating conditions in certain locations may still exceed product's certified performance limits. We noted a general tendency among contractors to only consider the stamp of technical approval when using a product, without necessarily considering its suitability for the project in question. Given two products with a technical approval, the less expensive one tended to be chosen regardless of capabilities.

## 4.5. Building Design Guides

The investigation into the building design guides found a level of detail and complexity too great to be easily manageable, causing a risk of some advice or recommendations being missed or for other reasons not followed. The nine building design guides were found to contain 322 paragraphs with a total of 977 individual recommendations. The design guides cross-reference each other, with the nine investigated guides referencing 22 other guides, presumably each containing around 100 more recommendations for the designer to consider. Additionally, as new guides are created or updated continuously, older guides are intended to be updated with new cross-references. However, there is a significant delay in this process. Both design guides that explicitly concerned vegetation on roofs were published too recently to be referenced in any of the other guides, which did not consider vegetated roofs at all.

On a detail level, the design guides explicitly consider technical risks. Technical risks were mentioned in a majority of the paragraphs. However, as stated by [64–66], information overload is a challenge involved wherever large amounts of information needs to be processed. The level of detail and amount of individual recommendations in the Building Design Guides makes it challenging to determine a hierarchy of risks and prioritize which aspects of the design to give the greatest level of attention. On a detail level, the recommendations give important advice, but a procedure for assessing the big picture appears to be missing. Risks and recommendations are sorted by topic and presented seemingly with equal importance, making it difficult to assess which challenges to give the highest priority in a situation with finite time and resources. While a skilled and experienced engineer can possibly manage this process, it is dependent on the experience of the individual designers and as such vulnerable to human error.

## 4.6. Project Delivery Methods

Because different project delivery methods organize the building process differently, each system allocates risks differently, and, therefore, the project delivery method should allocate the risk to the party with the greatest ability to understand it [67]. Consequently, the project delivery method will have a significant influence on how contractors perceive risks associated with blue-green roofs. In fact, the project delivery method is a variable affecting all actors as it determines, amongst others:

- The entry-point of the agent(s) (i.e., contractor participating in the design).
- The level of influence of the agent(s) (i.e., the contractor is responsible for the delivery).
- The level of integration (i.e., the use of a single project team to deliver both design and construction).

As seen, the introduction of concepts such as blue-green roofs make buildings more complex and, according to our interviewees, makes the building more prone to design flaws and construction errors [61]. A promising response to the challenges is the use of a collaborative project delivery method that seeks to align the client's interest with those of the supply chain [62]. For the contractor, collaborative project delivery proposes opportunities to reduce their quality risk (financial risk) as they

are involved in an earlier stage and are given the opportunity to collaborate closely with designers and the project owner [42,46]. Choosing the right project delivery method is, thus, important for the management of quality risk. A method that involves the contractor (and possibly sub-contractors and suppliers) in the design will provide a platform from which the contractor can contribute with their expertise in constructability.

## 5. Discussion

### 5.1. How Do the Various Actors in the Building Process Perceive the Quality Risks of Blue-Green Roofs?

As summarized in Table 1, each of the investigated actors own quality risk in a different way, but the general idea seems to be shared in common: "If a defect occurs, it should not be our fault". For actors on the delivery side of the construction process, it is highly important not to deliver a faulty piece of work that leads to a defect. For the project owner, it is important to receive a building without defects.

The public property owners interviewed in the first phase of the research highlighted the importance of a professional organization on the ownership side. Large property owners may have cooperation contracts with specific contractors and designers, which acts as a measure against fraudulent workmanship. It is also considered important for the owner to be present for quality control and inspection on the construction site.

There is a high level of trust between actors in the Norwegian construction industry [68]. During interviews, property owners expressed trust in the contractors to choose proper materials and solutions. This was also seen in the investigated design–build tenders, where most project owners gave contractors great freedom in selecting a design for the green roofs. The project owners express a general trust in the contractor to deliver a working product. The reason may be that the owners in smaller organizations do not have the necessary competence and resources to create detailed specifications. The chosen delivery model may greatly affect the responsibility and the influence the owner may have on quality control.

On the delivery side of the project organization, a primary concern common to all involved actors is to avoid the responsibility of eventual defects. Designers choose solutions based on design recommendations and reports by advisory bodies. A designer must not recommend a solution without knowing that it will work, with references to the proper documentation. Likewise, suppliers use standards and technical approvals to ensure that the performance limits of their materials are well documented. In theory, if all actors ensure they are not doing anything wrong, building defects ought not to happen. However, the means by which risk is managed by each actor do not necessarily overlap. In cases where the ownership of responsibility is unclear, risks may be ignored and cause defects to occur. An example in the case of blue-green roofs could be the uncertainty of whose responsibility it is to ensure the roof is cleared of all debris before installation of the blue-green layers begins.

### 5.2. How Are the Risks Associated with Blue-Green Roofs Currently Managed?

For project owners, the level of risk management appears to depend on the capabilities of the owner of the project in question, and the chosen project delivery model. Large building owners may have good relationships with trusted partners, employ technical experts to follow up the project, and inspect the construction site on a regular basis. These resources are not always available to every building owner, in which case the owner appears to place a great amount of trust in the contractors. Smaller actors were found to select design–build contracts more often, wherein the contractor is responsible for both design and construction of the building. However, this approach gives the owner less influence over the building and may affect risk management.

Designers anchor the solutions they recommend for the project by referring to general recommendations issued by building research organizations. In Norway, the SINTEF Building Design Guides are vital in this regard by containing recommendations on a detail level. However,

an assessment of the building design guides reveals a level of complexity that makes it challenging to see the overall picture and form a hierarchy of risks. In a sense, "the forest is lost among the trees." Making good use of the design guides in a practical situation requires a certain level of skill of the individual engineer.

Additionally, the building design guides do not cover every building detail or design feature. As an example, blue-green roofs are a novel building element not yet fully treated in the design guides. There is a guide for extensive sedum roofs, but there are challenges of blue-green roofs that it does not cover. For instance, the substrate layers of a blue-green roof will be thicker and have a greater capacity to hold water than an ordinary Sedum roof, which allows weeds to grow more easily. Maintenance plans based solely on recommendations for Sedum roofs may not cover this issue or other special cases. The engineers involved in the project will have to find their own solutions to such challenges. Thus, the risk management may rest entirely on the experience and expertise of individuals, which may not always be sufficient to meet the needs of every case. The sheer number of individual recommendations in the design guides also makes it challenging to use them to follow up work on the construction site. Third-party control of building physics solutions is required in construction projects in Norway, which serves to both reduce and share the risk for the designer.

Material suppliers manage quality risk by determining and documenting the usage and performance limits of their materials. In theory, this ensures that the suitability of a material for an individual application can be determined accurately. However, this approach to risk management does not necessarily prevent misuse of the material. Like with the building design guides mentioned above, the amount of information presents a level of complexity that may not be manageable in practice. While a skilled engineer working closely with a product could know the details of testing standards to know where and how to apply the product correctly, this information may not necessarily be available to the responsible person in the construction project. The required expertise exists somewhere, but one cannot assume everyone to be an expert. This sentiment echoes that of Josephson and Hammarlund [35], who found lack of knowledge to be one of the largest causes of defects in investigated construction projects.

### 5.3. How Can the Management of Quality Risk Be Improved?

The relevant quality risks in a project containing a blue-green roof exist within a manifold of partially overlapping perspectives and responsibilities, and the term carries different meanings to different actors, as shown in Table 1. The inherent complexity of the construction process means no single actor can take steps that reduce quality risks for everybody. Rather than suggesting measures for each involved actor, four different approaches to addressing quality risk management have been identified by the authors: (i) improvement of rules and regulations, (ii) improvement of competence, (iii) improvement of process flow, and (iv) improvement of best-practice design guidelines. Below, the merits and disadvantages of each strategy are outlined.

Rules and regulations vary between countries and sometimes even regions. In Norway, the regulations are function-based, which gives designers great freedom and leaves room for creativity and cross-disciplinary cooperation to test new solutions. The regulations already state that moisture damages must not happen [20], but leaves it up to the individual designers to find ways to achieve this. Further regulatory measures are likely to restrict the designer's freedom and may not fit within this style of regulatory framework. However, it could be possible to strengthen the requirements of documenting compliance with the regulations.

As demonstrated, competence and awareness of the main challenges are key to managing the risks of blue-green roofs. Several existing risk management strategies hinge on the competence of individuals. The required expertise to solve a problem may be found somewhere in the project organization, or somewhere in the documentation. However, not all project organizations are large enough to contain all the required expertise, and this research shows that documentation alone may not provide the right information in an understandable fashion to a non-expert. The choice of delivery

model to include better cooperation and sharing of knowledge may overcome these limitations to a certain degree.

A multi-disciplinary guideline to guide the process of acquiring blue-green roofs may be helpful as it could signal how to overcome challenges, set up design priorities, and allocate responsibility on a general basis instead of going deeply into the technical specifics. As an example, a report concerning the procurement process of expertise for the design and construction of climate adapted buildings illustrates how such a guideline could appear in practice [69]. The guideline would need to consider a manageable number of focus points to be practically useful to all involved actors.

For best-practice design guides, this article illustrates how they are considered helpful, but the volume of information is difficult to manage, and adding more recommendations might yield diminishing returns. It is evident that a structuring of the recommendations might be more useful than solely making more recommendations. A motivating example of this structuring was carried out by Asphaug et al. [70], who assessed the design recommendations for habitable basements in the SINTEF Design Guides and extracted 10 key challenges around which the recommendations were grouped. Creating such a hierarchy by allocating the hundreds of individual recommendations to a manageable number of main challenges makes it easier to assess risks in a systematic fashion.

## 6. Conclusions

The research shows that the different actors in a construction project perceive and manage quality risks differently. Each actor generally has a means to manage quality risk in their own part of the process, which in theory creates an overlapping patchwork of risk management that in sum will cover the entire project. However, there are gaps between the various risk management strategies, usually where the responsibility of one actor ends and that of another begins. Notably, it is shown that documentation meant to avoid risks will either be insufficient to cover every detail, or too complex to be put to practical use in every project. A significant level of expertise is required to create a whole picture from the many details presented in the best-practice design guidelines and to understand how to adapt and apply them in special cases. This makes them insufficient as a sole tool for risk management. Likewise, the technical approvals of construction products may not necessarily suffice to assess their suitability in a given project, because the details of their performance limits are not easily understandable to the procurement agent of the project. Project owners may subvert the need for detailed expertise in the building process by selecting contract forms that place the responsibility for design and construction on a single contractor; however, they will still end up carrying the risks of defects in the long term.

It follows that a framework to reduce the total quality risk in a building project cannot focus solely on one actor or one phase of the process, but it needs to consider multiple perspectives and project phases. It also needs to address the issue of thoroughness versus complexity, hitting a balance of covering enough issues without being too voluminous to be practical and understandable to those who use it—which is to say, anyone in the building process.

Future work will attempt to analyse the quality risks associated with blue-green roofs in a systematic fashion and present a framework for risk management from multiple perspectives. An analysis of the situation of quality risk for blue-green roofs in other countries could also be conducted. The overall goal will be to reduce the overall quality risk associated with blue-green roofs, delivering a reliable means of stormwater management without compromising the integrity of the building.

**Author Contributions:** Conceptualization, E.A., B.T., J.L. and T.K.; methodology, E.A., A.E., B.T., J.L. and T.K.; validation, B.T. and T.K.; formal analysis, E.A., A.E., B.T. and O.T.; investigation, E.A. and A.E.; resources, B.T. and T.K.; data curation, E.A. and A.E.; writing—original draft preparation, E.A.; writing—review and editing, E.A.; visualizations, E.A., B.T. and T.K.; supervision, B.T., J.L. and T.K.; funding acquisition, B.T. and T.K. All authors have read and agreed to the published version of the manuscript.

**Funding:** This research was funded by the Research Council of Norway, grant number 237859.

**Acknowledgments:** This article is based on research previously presented in conference papers [42,53–55] and a technical working document [56]. While each of the previous papers regarded one issue or perspective separately, this work is meant to tie them together and provide an overall picture of risk management in the building process of blue-green roofs. The authors gratefully acknowledge the financial support by the Research Council of Norway and several partners through the Centre of Research-based Innovation "Klima 2050" We would like to extend a special thanks to CAD operator Remy Eik.

**Conflicts of Interest:** The authors declare no conflict of interest.

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
