# Peer review of "Perspectives on Quality Risk in the Building Process of Blue-Green Roofs in Norway"

_buildings, doi:10.3390/buildings10100189_

Round 1

Reviewer 1 Report

I attach a pdf file.

Author Response

Thank you for a thorough review. Please see the attached document for details.

Reviewer 2 Report

General notes:

The manuscript “Perspectives on quality risk in the building process of blue-green roofs”, investigates how different actors perceive and manage quality risks related to blue-green roofs, through document studies and interviews with actors in the Norwegian building sector. Results showed that different actors perceive and manage quality risks differently and that reducing the risks in the activities of one actor may not necessarily reduce the risk of defects in roofs.

The paper has the potential to be published in Buildings, after some minor modifications. In particular, it should be highlighted that the study focuses on the Norwegian system and conditions and the requirements to extend this approach worldwide should be included.

 The manuscript is overall clear; however, the punctuation needs to be carefully revised (capital letters after column should be removed) and the text would benefit from a native speaker revision to improve the English quality.

Title

This work can be considered as a regional study: climatological and socioeconomical conditions are specific of Norway and the conclusions derived from this study can not be automatically extended to a wider context. I would recommend highlighting this in the title, adding “in Norway” to it.

Specific notes:

[1, 36-37] Please add some references to justify the different purposes of green roofs and blue-green roofs. The difference between these two structures is that the blue-green roofs presents both retention and detention capacity, while traditional green roofs rely mainly on the retention capacity. Consequently, the structures can be used for different purposes. Please consider adding this concept.

[2, 44-45] Please add some reference to justify this sentence

[Section 2.1] Consider to include the following references for a more complete and detailed description of blue-green roofs

Shafique M, Kim R, Lee D. The potential of green-blue roof to manage storm water in urban areas. Nature Environment and Pollution Technology 2016a; 15: 715.

Shafique M, Lee D, Kim R. A field study to evaluate runoff quantity from blue roof and green blue roof in an urban area. International Journal of Control and Automation 2016b; 9: 59-68.

[5, Figure 2] It’s not clear why “construction organization” has 2 rectangles and what the circles represent. Please explain better the meaning in the legend.

[Results] Although the obtained results are qualitative a summarizing table/scheme could be beneficial for the “Results” and “Discussion” sections.

[Conclusion]: Consider including in the future works an evaluation of this study in other countries, with the aim to derive conclusions valid worldwide.

Author Response

Thank you for a thorough review. Please see the attached document for detailed response.
